# Evolution of Vaccines Formulation to Tackle the Challenge of Anti-Microbial Resistant Pathogens

**DOI:** 10.3390/ijms241512054

**Published:** 2023-07-27

**Authors:** Francesco Tognetti, Massimiliano Biagini, Maxime Denis, Francesco Berti, Domenico Maione, Daniela Stranges

**Affiliations:** 1Department of Pharmaceutical and Pharmacological Sciences, University of Padova, Via F. Marzolo 5, 35131 Padua, Italy; 2GSK, 53100 Siena, Italy

**Keywords:** vaccine, AMR, formulation, antigen, adjuvant

## Abstract

The increasing diffusion of antimicrobial resistance (AMR) across more and more bacterial species emphasizes the urgency of identifying innovative treatment strategies to counter its diffusion. Pathogen infection prevention is among the most effective strategies to prevent the spread of both disease and AMR. Since their discovery, vaccines have been the strongest prophylactic weapon against infectious diseases, with a multitude of different antigen types and formulative strategies developed over more than a century to protect populations from different pathogens. In this review, we review the main characteristics of vaccine formulations in use and under development against AMR pathogens, focusing on the importance of administering multiple antigens where possible, and the challenges associated with their development and production. The most relevant antigen classes and adjuvant systems are described, highlighting their mechanisms of action and presenting examples of their use in clinical trials against AMR. We also present an overview of the analytical and formulative strategies for multivalent vaccines, in which we discuss the complexities associated with mixing multiple components in a single formulation. This review emphasizes the importance of combining existing knowledge with advanced technologies within a Quality by Design development framework to efficiently develop vaccines against AMR pathogens.

## 1. Introduction

The discovery of chemically synthetised antimicrobials in the 19th century was a milestone in the field of medical research, and in 1928, Alexander Fleming introduced the world to a new era of bacterial disease treatment when he identified the first non-toxic antibiotic (i.e., Penicillin G). Since then, the combination of antimicrobials and antibiotics has contributed to the prolongment of human life expectancy, and people in high-income countries can now expect to live to 81 years of age [1]. Despite the outstanding impact his discoveries had in medicine and on the quality of human life, Fleming in 1946 said: “There is probably no chemotherapeutic drug to which in suitable circumstances the bacteria cannot react by in some way acquiring fastness [resistance]” [2]. This ominous prediction is now being realised, as resistance to antimicrobials (antimicrobial resistance; AMR) is now recognised as future potential health emergencies [3]. The mechanisms through which bacteria develop AMR can be defined as either intrinsic to the bacteria or acquired. The main pathways through which resistance is obtained by bacteria are (i) hydrolysis of the antibiotic structure; (ii) expression on the pathogen membrane of multi-drug resistance efflux pump systems; and (iii) the mutation of antibiotic targets [4]. The acquired mechanisms derive from the capacity of bacteria to transfer genes via plasmids, transposons, bacteriophages or other types of genetic material, enabling the acquisition of new resistance mechanisms [5,6,7]. Nowadays, the spread of AMR is increasing due to a variety of factors, including inadequate regulation of antibiotics usage and the misuse of therapeutics and their use as livestock growth promoters. The number of deaths related to AMR in 2050 is estimated to overtake those due to cancer, and the economic impact related to the increased frequency of ineffective treatments resulting in a longer hospital stay and loss of global production will be about will be about 1.5 billion euros per years in the EU alone [8,9]. A list recently published by the World Health Organisation (WHO), highlighted “ESKAPE” (*Enterococcus faecium*, *Staphylococcus aureus*, *Klebsiella pneumoniae*, *Acineto-bacter baumannii*, *Pseudomonas aeruginosa*, and *Enterobacter* spp.) as the highest priority AMR pathogens [10]. There is now global recognition of the need to tackle drug-resistant infections with a combined effort. As defined by world-leading specialists in the field, the first steps against AMR pathogens expansion are as follows: (i) a collective awareness campaign; (ii) the improvement of hygiene and prevention of the spread of infection; (iii) the reduction of antibiotic use in agriculture; and (iv) improving the global surveillance of antimicrobial drug consumption [11]. The rapid expansion of AMR to include newer classes of antibiotics and the difficulties associated with the discovery of new therapeutic molecules render vaccination one of the most promising strategies for the control of AMR expansion [12]. Antibiotics are prescribed to already infected patients after pathogen proliferation, causing a delay between the infection and the therapeutic effect. Antibiotics can also cause damage to bystander bacteria and microbial flora, which creates a selective environment that favours AMR bacteria proliferation and the selection of new drug-resistant mutant strains [13,14]. In contrast, vaccines prevent the proliferation of the pathogen by inducing a protective immune response prior to encountering the pathogen. Moreover, thanks to the phenomenon of herd immunity, they shield unvaccinated individuals and therefore significantly mitigate the prevalence of disease across the entire population [8]. The indirect protection derived from diffused vaccination is critical to protect fragile patients with immunosuppressive diseases and others who are unable to receive vaccines. The WHO has identified 12 families of antimicrobial-resistant pathogens with three different levels of priority: critical, high, and medium [15]. A second classification divides the pathogens into four groups depending on the availability of vaccines to treat them: Group A have already-licensed vaccines available; Group B have vaccines that are in late-stage clinical trials; Group C have vaccine candidates in early stage clinical trials; and Group D have only a few candidate vaccines at the clinical stage or none at all [15]. In 2018, an expert pool on behalf of the Global Alliance for Vaccines and Immunisation (GAVI-The Vaccines Alliance) defined the selection criteria for vaccines against AMR pathogens: reduction of mortality, prevention of use of antibiotics, morbidity, sense of urgency due to treatment impact, and ethical importance [16]. There are numerous challenges in the formulation of new vaccines against AMR pathogens. A plethora of technologies are available in anti-bacterial vaccine development: live–attenuated or killed microorganisms, recombinant proteins, mRNA-based vaccines, reverse vaccinology applied to Generalised Modules for Membrane Antigens (GMMAs), glycoconjugates, and nanoparticles [8].

This review will focus on the main aspects related to all stages of the formulation, development, and characterisation of new vaccines for AMR pathogens, highlighting where possible the most relevant industrial approaches currently used.

## 2. State-of-the-Art and Innovative Vaccines Formulation Strategies to Prevent AMR Pathogen-Induced Diseases

### 2.1. Formulation of Vaccines against AMR Pathogens

The continuous improvement of current technologies is fundamental for the development and manufacture of safer and more effective vaccines against a wider spectrum of pathogens [17]. In this chapter, we will review the main characteristics of new antigen technologies and their use against AMR pathogens approved vaccines or in vaccines currently under development. Figure 1 provides a schematic representation of the vaccines platforms and adjuvants reviewed in this manuscript.

The aim of vaccination is to achieve an effective immune response in the target population, thus preventing the disease or decreasing its severity. Vaccine efficacy is strongly related to the potency, stability and long-term shelf-life and affected by the quality of manufacturing processes [18]. To achieve the optimum values for these metrics, the antigen must be administered as part of a formulation with other components with different functions, such as excipients and adjuvants. A vaccine formulation can be described as a complex biopharmaceutical product containing a mixture of active components which is stable under a wide range of storage conditions and assembled so as to obtain a product that is reproducible, scalable, and robust [19]. Vaccines against AMR pathogens are formulated with consideration for the specific need to induce protection against multiple isolates or one or more species. Formulations containing different antigens to deliver an effective product against the highest number of circulating bacteria strains are therefore used wherever possible. The development of a vaccine that contains numerous active components challenges formulators to define the formulation conditions in which the stability of all the formulated components, and the compatibility between them, is guaranteed.

Due to the low immunogenetic response typically delivered by most highly purified antigens when administered alone, adjuvants are often added to vaccine formulations, to increase and modulate the resulting immunogenic profile [19].

Another important aspect of developing new vaccines against the diffusion of antimicrobial resistant bacterial disease is their rapid approval. A new silent pandemic concerning the rapid diffusion of new mutant antimicrobial resistant pathogens is a growing problem. Each year, 4.95 million deaths are associated with AMR, and 1.27 million deaths are directly attributed to it [20]. Rapid development of vaccines is possible as demonstrated with the SARS-COVID-19 pandemic, and prior knowledge of mRNA technology together with the requisite support from government agencies would allow for the rapid approval of a new vaccine technology platform within a pandemic setting [21].

### 2.2. Recombinant Protein-Based Antigens for Bacterial Targets

#### 2.2.1. Generalities of Protein-Based Anti-Bacterial Vaccines

Proteins are one of the most common antigen classes used in vaccination. Proteins can be naturally expressed, synthetic, or recombinant in origin [22]. Recombinant proteins are expressed by host cells such as *E.coli* and mammalian, yeast, or insect cells [23,24,25], and are produced starting with the introduction of a genetically modified plasmid which encodes the information to express the desired protein inside the host cell [26,27]. In recent years, huge improvements in the field of genome manipulation have taken place, meaning it is now easier and faster to produce more diverse heterologous proteins, enabling the production of multi-antigen vaccines offering broader protection [28]. The topic of recombinant protein expression and purification has been reviewed extensively in Varsha Gupta et al.’s review ”Production of Recombinant Pharmaceutical Proteins” [29].

To induce the broadest immune response possible in a given vaccine, it is essential to select the correct protein antigen to be administered. Bacterial surface proteins or toxins are commonly selected as they are characterised by thoroughly exposing the immune system to the bacterial antigens. Antibodies produced against surface proteins typically trigger opsonophagocytosis by macrophages and neutrophils against the pathogen. Alternatively, the antibodies produced against toxins block its virulence-related mechanisms and its toxicity [30]. Moreover, to increase efficacy against AMR pathogens, it is important to identify proteins that are highly conserved among the different bacterial isolates [31].

There are fewer risks associated with the use of recombinant protein antigens as compared to natural purified proteins, as they circumvent issues such as the co-purification of contaminants, the reconversion of toxoids to their toxigenic form, or the difficulties in obtaining satisfactory amounts of purified antigen [32].

The efficacy of recombinant protein-based vaccines is strongly related to the preservation of their structure over time, from the point of manufacture to the administration to recipients. The stability profile is therefore a critical consideration, and suitable excipients are typically included in the final drug product formulation to ensure its stability. For example, the most appropriate pH and buffer are selected to ensure that the antigen preserves its native secondary and tertiary structure and to avoid neutralisation of the protein surface charge, as this can trigger aggregation [33]. In fact, protein aggregation is directly dependent on protein colloidal stabilisation. For proteins in aqueous solutions, it is important to maintain a pH far from the isoelectric point, thus avoiding surface neutralisation and favouring electrostatic repulsion. The choice of the most appropriate buffer for a given system depends on its buffering range capacity, which must be compatible with the stable pH range of the protein. Other excipients may be introduced to the vaccine formulation to further improve antigen stability, improve product manufacturability, or improve product stability (in both liquid and/or lyophilised form). The most-used excipient classes are amino acids and carbohydrates, and surfactants can also be added. Surfactants increase protein stability in two ways: by positioning themselves on protein surface interfaces, they preclude interaction with manufacturing materials as well as with each other [34]. Carbohydrates are the most commonly used protein formulation stabilisers for lyophilised product. During freeze-drying, excipients such as sucrose and trehalose generate a solid matrix that, below the glass transition temperature, is both amorphous and extremely viscous. Moreover, the presence of multiple hydroxyl groups within these excipients further stabilises the protein via hydrogen bonding with surface sites on the protein (after water evaporation) [35].

#### 2.2.2. Applications to AMR Pathogens

There are numerous examples of recombinant proteins that have been shown to deliver effective protection in recipients in both clinical and pre-clinical trials against AMR pathogens.

Preclinical studies on a YidR protein antigen for the development of a vaccine against *Klebsiella pneumoniae* infections demonstrated that when mice were infected with *K.pneumoniae*, vaccination with the YidR vaccine resulted in a survival rate higher than 90% (as compared to 0% in the control group) [27]. More recently, a phase I clinical trial tested the efficacy of a multi-component vaccine against *S.aureus* to stimulate a robust humoral response in recipients. The vaccine was formulated with staphylococcal protein A (SpA), α-haemolysin (Hla), iron surface-determinant B N2 domain (IsdB-N2), staphylococcal enterotoxin B (SEB), and manganese transport protein C (MntC) antigens and induced a robust and efficient immune response in adult patients [36]. A new recombinant protein vaccine, containing an F2 antigen targeting *Chlostroides difficile,* is currently under investigation in a phase 1 clinical trial [15].

Despite the promise of these recombinant protein antigen candidates (all of which are still under development), this class of vaccine is usually characterised by low immunogenicity, which renders the use of immuno-enhancers (i.e., adjuvants) necessary in the final drug product formulation [32]. There also exist alternative strategies to enhance the immunogenicity of recombinant protein vaccines (for example, as part of a nanosystem matrix), which could remove the need for adjuvants [37]. This is an active field of research and will be discussed in more detail in the coming sections.

### 2.3. Membrane Vesicle-Based Antigens

#### 2.3.1. Generalities on Membrane Vesicles

Both Gram-positive and Gram-negative bacteria can form structures known as membrane vesicles (MVs), which are bilayer spheres with a diameter in the range of 20–250 nm, and they possess a large variety of microbial-associated molecular patterns (MAMPs) [38]. Pathogen recognition receptors (PRRs) located on both immune and epithelial cells recognise MAMPs, leading to the activation of innate immune responses, and an initial defence mechanism against invading pathogens in host organisms [39,40]. Outer membrane vesicles (OMVs) are bilayer phospholipidic structures naturally generated by gram-negative bacteria, with an outer layer composed of lipopolysaccharides (LPSs), membrane proteins, and receptors. Internally, OMVs possess a thin layer of peptidoglycan and contain periplasmic proteins as well as nucleic acids [41]. These structures have been considered attractive targets for vaccine development as they expose the immune system to bacteria, such as membrane structures.

Recent advancements have allowed for the genetic modification of a bacterial genome to produce bacterial strains characterised by an over-vesiculating phenotype. GMMAs are OMVs produced by genetically modified bacterial strains to obtain weaker bonding between the outer membrane and the peptidoglycan of the bacteria [42,43]. This weakened bonding is due to the deletion of the *TolR* gene and results in a more pronounced shedding process. Furthermore, whilst the production of OMVs is currently based on detergent extraction from wild-type Gram-negative bacteria, the production of purified GMMAs is based on a rapid downstream filtration process, which leads to comparatively higher yields and lower production costs [44,45].

Because both OMVs and GMMAs express lipopolysaccharides on their surface, they are reactogenic. To control the inflammatory response activated by the LPS, bacteria are further genetically modified to reduce the degree of lipid A phosphorylation and acylation, thus limiting the activation of the TLR4 receptor and preserving their intrinsic adjuvanticity [42,46,47]. In fact, because LPSs are recognised by TOLL-like receptor 4 (TLR4) and TOLL-like receptor 2 (TLR2), GMMAs can induce the release of IL-1β by stimulation of caspase-1 [48]. Another component expressed (in *Salmonella* GMMAs) is flagellin, which can stimulate the production of cytokines, including TNF-α and interleukin 6. It is therefore critical to modify the LPS content in order to modulate reactogenicity while maintaining immunogenicity [47]. These modifications are encoded within the lipid structure via the introduction of mutations in the gene which encode for acyltransferases such as *HtrB* or *MsbB* in *Shigella MsbB* or *PagP* in *Salmonella,* and *LpxL1* in *N. meningitidis* [49].

The size of MVs, moreover, enables them to diffuse through the lymphatic system or be phagocyted by APCs. Their recognition by APCs activates CD8+ T cells, dendritic cells (DCs) antibody production by B-cells and macrophages few hours post-vaccination [50,51].

Because of their adjuvant activity (and improved immunogenicity when compared with conventional carrier proteins), membrane vesicles could also potentially be used as carriers for heterologous antigens [52,53].

GMMA and OMVs are typically formulated with an aluminium adjuvant, which adsorbs the membrane vesicles onto its surface in the final drug product formulation. To further enhance GMMA tolerability, Shigella *sonnei*-based vaccine was adsorbed on aluminium hydroxide and demonstrated to lower GMMAs’ pyrogenic response in rabbits [54]. Moreover, aluminium-adsorbed *S. typhirium* and *S. enteritidis* GMMA antigens have been demonstrated to have an improved thermal stability profile with respect to non-adsorbed antigens [55]. This could make possible a reduction in the reliance on cold-chain storage and transport, which is typically required and limits vaccine distribution in low-income countries. Recent studies have introduced the possibility of using GMMAs as a carrier for multiple antigens contemporaneously, producing an effective immune response in mouse models without inducing immune interference [56].

The multivalent potential of this technology suggests that GMMAs could be a promising instrument to develop new vaccines against AMR pathogens by immunising recipients against multiple bacteria strains with a single vaccine.

Genetic engineering of *E. coli* bacteria to promote the expression of heterologous proteins within the OMV membrane could be a promising strategy to enhance the efficacy and activity of OMVs. In fact, protein expression on the surface of OMVs increases the immunogenicity of administered antigens that are not sufficiently immunogenic when administered in their purified form. This is possible as *E. coli* expresses transmembrane pore-forming proteins called cytosolin A (ClyA), which spontaneously migrate to the membrane of the OMV. When bacteria are genetically modified to express protein antigens bonded to the C terminus of ClyA, the integration of the pore-forming protein in the OMV membrane favours the exposure of the antigen on the surface of the vesicle [57]. The efficacy of an engineered *E. coli* OMV vaccine was shown against *Streptococcus pneumoniae* challenges in mice. The *Streptococcus pneumoniae* PspA antigen was expressed in lumen of *S. enterica*, and OMVs demonstrated protection in mice against a challenge with S. pneumoniae, while control group (which received purified PspA) was not protected [58].

An effective method to induce multivalent antigen expression on the surface of an OMV is by manufacturing glycoengineered OMVs. These are nanoparticles obtained from the introduction of a genetic locus that encodes for the glycan structure of interest, which is not naturally expressed by the bacteria. The glycan, which is initially exposed on the cytosolic face of the inner membrane, is flipped on the periplasmic side by the Wzy flippase protein and polymerised to yield a higher molecular weight chain [59]. The glycan is transferred to the lipid A by the WaaL ligase, and the customised lipopolysaccharide molecule is then exported onto the bacterial surface using the Lpt protein complex [59,60].

Gram-negative OMVs are currently more widely used within the field of vaccine development, though Gram-positive MVs are of growing interest. The release of MVs was demonstrated in a large variety of Gram-positive pathogens, such as *S. aureus*, *S. pneumoniae*, and *M. tuberculosis* [61]. Due to the greater complexity of their bacterial cell wall, their generation and release mechanism are still debated [61,62]. As with OMVs, a large number of proteins are exposed over Gram-positive MV surfaces, and more than 90 vesicular proteins have been discovered in proteomics studies performed on *S. aureus* MVs [63].

#### 2.3.2. Application to AMR Pathogens

MV-based vaccines are currently used and tested in developing vaccines against different AMR pathogens with effective results.

For example, a MeNZB OMV-based vaccine developed to control the outbreak of *N. meningitidis* B was approved in New Zeland between 2004 and 2008. Another example of a vaccine approved for *N. meningitidis* containing OMVs is the 4CMenB vaccine (Bexsero, GSK). It is a four-component vaccine containing three recombinant surface-exposed protein antigens and an OMV from the New Zealand strain NZ98/254 [64]. *Shigella* is another example of an AMR pathogen target against which MV technology could be exploited in vaccine development. A GMMA-formulated O-antigen-based vaccine, a candidate against *Shigella sonnei,* has been tested in phase 1 and 2 clinical trials in adults [45,65]. Both the safety of the formulation and immunogenicity against antigens were demonstrated, and good serum bactericidal activity was reported in patients that were treated with a booster dose administered 2–3 days afterwards [66]. Despite promising phase 1 results, in phase 2 trials a lack of efficacy against shigellosis was found. This was attributed to an insufficient degree of immunogenicity, suggesting that an increased OAg dose and a greater time interval between the first and the second immunisation may be necessary [45,65]. More recently, a four component GMMA-based vaccine against *Shigella* has just started a phase 1/2 clinical trial to evaluate its safety and immunogenicity.

A promising application of Gram-positive EVs is the potential to develop a vaccine against AMR pathogen *S. aureus*, for which there is currently no approved vaccine. A preclinical study reported the isolation of MVs from *S. aureus* USA300 overexpressing HlaH35L and LukE, from which the authors were able to induce immunogenicity and protect mice in an *S. aureus* lethal sepsis model [62].

These examples demonstrate the possibility of using MVs as an effective platform to obtain multivalent vaccines against multiple AMR pathogens more quickly than via traditional methods. Such a technology may be useful in future pandemic situations.

### 2.4. Glycoconjugate-Based Antigens

#### 2.4.1. Generalities on Glycoconjugates

Glycoconjugation in vaccines refers to the covalent linking of bacterial oligo- or polysaccharides (PSs) to proteins. In antimicrobial vaccines, the use of bacterial polysaccharide antigens alone triggers the activation of the immune system via a T-cell independent mechanism, and as such the immune system is unable to produce memory B-cells. This is particularly the case in infants, young children (under two years of age), the elderly, and immunocompromised patients. Moreover, the protection provided by the vaccine is also often insufficient in adults, as it induces only short-lasting antibody responses [67].

A widely used strategy to overcome the lack of T-cell response to PS antigen-based vaccines is via conjugation of the antigenic PS to an immunogenic carrier protein, thereby forming a glycoconjugate vaccine. Such a system enables the activation of the T-cell-dependent response in addition to the production of memory B-cells, thus inducing a good immune response even in children under two years old and in elderly people. The most common examples of proteins used for conjugation to polysaccharides (PS) are tetanus toxoid (TT), diphtheria toxoid (DT), and a non-toxic mutant form of DT (CRM_197_) [1,58,59].

The manufacturing process of glycoconjugate vaccines encompasses the separate production of the carrier protein and antigens, which are subsequently conjugated via chemical reactions. This process is long and requires a high number of control steps [68,69].

One of the most important requirements in the production of glycoconjugate-based vaccine production is the need to stabilise the carrier protein structure to avoid its denaturation, which may in turn cause its physical degradation or aggregation. The typical formulative approach employed is similar to the one described for recombinant protein vaccines, with the choice of buffer and the introduction of surfactants and stabilisers being key to providing structural and conformational stability of the antigen. Additional considerations should be taken into account regarding the stability of the chemical structure of the polysaccharide, with particular attention paid to the degree of O-acetylation of the PS, which affects epitope arrangement and is highly sensitive to the pH of the formulation. Polysaccharide chain length should also be carefully monitored in addition to the level of conjugation to the carrier protein to minimise the degree of free (unconjugated) saccharide. These attributes should be holistically evaluated during pre-formulation studies in order to define a suitable formulation space, as illustrated in Figure 2.

Improvements in the field of glycoconjugate methodology are leading to the development of more effective and novel vaccines, for example against mucosal or intracellular pathogens or against AMR pathogens constituted from multiple serotypes [70]. A new platform technology known as the multiple antigen-presenting system (MAPS) is characterised by an affinity-based coupling approach between bacterial PS and proteins, to mimic the chemical and physical features of a whole-cell construct [70]. The first step in the process is to individually biotinylate the target PSs, whilst separately genetically fusing the target protein antigen to a biotin-binding protein, rhizavidin (rhavi). After the incubation of the rhavi-fused antigens with biotinylated PSs, spontaneous formation of macromolecular complexes occurs. Because the coupling is independent of antigen properties, the functionalisation of the resulting MAPS is highly modulable and versatile and allows for the introduction of multiple protein antigens, fusion proteins, and PSs within the same construct [63]. This feature of the MAPS platform could be useful in the development of vaccines against AMR pathogens, as these pathogens (e.g., *S. pneumoniae Salmonella*) typically have many serotypes, meaning a broad spectrum of protection is required. This is critical for effective immunisation against the greatest number of pathogen serotypes possible, to thus reduce the spread of new mutations.

Bioconjugation is a single-step process based on the in vivo synthesis of glycoconjugates, exploiting engineered *Escherichia coli* bacterial strains [71]. The bacteria co-express heterologous polysaccharide antigens linked to the carrier protein carrying glycosylation sites. The bacterial enzyme Pg1B forms a link between the carrier protein and the glycan, preserving the morphology of the antigenic components. The bioconjugation process happens entirely inside the bacterium, which increases process reproducibility and significantly reduces manufacturing complexity [72].

However, bacterial PSs are often complex and composed of a mixture of different monomers linked with different stereochemistry, and therefore a precise control of the glycosylation profile requires the presence and coordinated control of many enzymes.

#### 2.4.2. Glycoconjugate Vaccines Application to AMR Pathogens

Glycoconjugate-based vaccines are already approved for many child vaccines sold by several manufacturers. This includes the glycoconjugate vaccine for use against meningococcal serogroups A, C, W-135, and Y (Menveo, GSK (Siena, Italy)) [73]. Pneumococcal disease is another common target for traditionally produced glycoconjugate vaccines, and several multi-valent (containing up to 23 antigens) pneumococcal conjugate vaccines are currently commercially available or in active clinical trials [15]. The vaccine administered depends on the age of the recipient: children under two years old receive a double dose of the pneumococcal conjugate vaccine (Prevnar 13; PCV). High-risk people and the elderly receive the pneumococcal polysaccharide vaccine (Pneumovax-23; PPV) [74]. In 2021, a new pneumococcal vaccine, Prevnar20, was approved for use in adults aged 18 and older [75].

A promising perspective in the formulation of new vaccines against a wide variety of AMR pathogens is given by the new MAPS. This system has been used in several clinical trials. A pneumococcal capsular polysaccharide (CPS)-14 formulation generated high antibody titres in immunised animals with a low antigen dose [70]. Evidence of strong Th1 and Th17 activation was detected, which may have been due to the larger size of the MAPS complex in comparison to free PSs or classical glycoconjugates, thus inducing secretion of proinflammatory cytokines [70]. A vaccine against both *Salmonella typhi* Vi and *paratyphi* A have been tested in different animal models, with the studies confirming immunogenicity and induction of affine antibodies and polysaccharide-specific memory B-cells [76]. Human trials are currently ongoing for both healthy (i.e., 18–65 years) and elderly (i.e., 65–85 years) adults using a novel 24-valent pneumococcal vaccine (ASP3772). The safety of the vaccine was assessed at different dosages, and the immune response was comparable to or higher than that obtained with the commercial Prevnar13 vaccine [77].

Vaccines formed via bioconjugation have already been tested in clinical trials against *E. coli* infections, and an acceptable safety profile and a good vaccine-specific immune response were found [78,79]. Additionally, a new quadrivalent *Shigella* bioconjugate vaccine carrying *Shigella flexneri* serotypes 2a, 3a and 6 and *Shigella sonnei*, is currently being tested in a phase 1/2 dose-finding, multi-age (adults–children–infants) study in Kenya [72].

Innovative technologies such as the MAPS will help us develop vaccines to fight AMR on two fronts: speed and breadth. By being inherently multivalent, the MAPS allows for the combination of up to 30 antigens for an increased coverage of strains and serotypes. Moreover, being a versatile and modulable platform, it will be easily redeployable against other pathogens with reduced development time. As these traits are inherent to the technology, once it becomes established within the field, the MAPS should therefore improve vaccine development against AMR (and other pathogens) both in terms of speed of production and breadth of coverage.

### 2.5. RNA-Based Vaccines

The approval of mRNA vaccines against the SARS-Cov2 virus introduced a new paradigm in vaccination and paved the way for new possible applications of this technology towards a multitude of other treatments. The mechanism of action of mRNA vaccines is based on the internalisation of mRNA, which contains the antigen coding sequence within cells to induce them to express the encoded protein and present it to the immune system. An mRNA sequence is typically composed of an open reading frame which encodes the target antigen and is flanked by untranslated regions (UTRs), a five-prime (5′) cap, and a terminal poly (A) tail [80]. Naked mRNA has been shown to degrade rapidly inside the body via a large variety of mechanisms. It is therefore necessary to increase the stability of mRNA-based vaccines as part of their formulation strategy. To this end, lipid nanoparticles (LNPs) have been demonstrated to be very effective delivery vehicles and are currently the most widely used systems for the encapsulation and delivery of mRNA vaccines. LNPs are typically obtained by mixing different ratios of four components: an amino lipid; a phospholipid; cholesterol; and a PEG-ylated lipid [81]. The use of ionisable amino lipids reduces the interactions between blood stream cell membranes as compared to the cationic form. When LNPs are internalised in endosomes, the acidic environment leads to amino lipid protonation, facilitating the breaking of the endosome membrane and endosomal escape. The other lipid components within the LNP formulation are used to improve various attributes of the LNPs, such as their stability, delivery efficacy, tolerability, and biodistribution. Cholesterol has been shown to modulate the rigidity of the membrane, leading to an overall increase in the stability of the structure. Phospholipids have several different roles within the LNP, including improving encapsulation and cellular delivery. PEG-ylated lipids are used to modify zeta potential and particle size, prolonging the time that LNPs can circulate within blood and reducing their rate of clearance [82]. mRNA can be successfully and reproducibly encapsulated within LNPs via rapid mixing of an alcoholic solution of lipids and an aqueous solution of mRNA at pH lower than the pKa of the ionisable lipid mixture. This stable structure is a result of electrostatic interactions between the mRNA molecules and the amino lipids, which allows for the mRNA to be encapsulated within the core of the LNP structure [82].

The main current limitation of using this technology in the context of bacterial vaccination is the risk of inducing modifications to the bacterial antigen genetic sequence when expressed in eukaryotic systems. This step occurs naturally for viral proteins [83].

Despite this challenge, some promising examples exploiting mRNA in the development of vaccines against bacterial diseases are here described.

Self-amplifying mRNA (SAM) modality was used to express antigens from Group A (GAS) and Group B (GBS) streptococci. It was demonstrated that the eukaryotic cells transfected with SAM produced proteins similar to the prokaryotic-expressed antigens. Both the vaccines induced a strong immune response and were shown to be partially protective in mice via passive and active immunisation. This was the first time an mRNA vaccine was demonstrated to be protective against bacteria in preclinical animal models of infection [84]. More recently, an mRNA LNP vaccine has been tested in preclinical models against Gram-negative bacterium *Yersinia pestis,* which is a target of great importance as two of its strains have demonstrated resistance to antibiotics. The vaccine is based on an mRNA strain modified to express monomeric *caf1* capsule antigen coupled with a mammalian signal peptide (SP) sequence, which replaces the native bacterial signal sequence. The mRNA sequence was also enriched with guanine and cytosine (GC) tandems that were demonstrated to increase protein expression and optimised to improve its stability. The candidate vaccine demonstrated high levels of both humoral response and protection against *Y. pestis* infection in mouse models [83].

A second study investigated the efficacy of an mRNA LNP-based vaccine encoding for two highly conserved proteins: the *P. aeruginosa* (PA) V-antigen (PcrV) and a fusion protein of membrane porin F (OprF) and lipoprotein l (Oprl) from *P. aeruginosa.* Both mRNA-PcrV-LNP and mRNA-OprF-I-LNP were demonstrated to be able to induce high antibody titre levels. Moreover, the immune response obtained from the administration of mixed mRNA-PcrV-LNP and mRNA-OprF-I-LNP was higher than that obtained from protein vaccines. Notably, both the mRNA against *P. aeruginosa* was demonstrated to be protective against PA infection [85].

Another example of a study of an anti-bacterial mRNA vaccine was recently published by Rhea N C. et al.: a novel self-replicating RNA platform formulated in a nanostructured lipid carrier was tested with the hypothesis that a longer antigen exposure would result in enhanced immune response against selected *Mycobacterium tuberculosis* vaccine candidate antigens. Moreover, the study compared homologous ID91 *Mycobacterium t.* protein or replicating RNA (repRNA) vaccination with heterologous RNA-prime, protein-boosting, or combination immunisations. The experimental results highlighted the fact that the greatest reduction in bacterial burden and a unique humoral and cellular immune response profile was obtained when heterologous immunisation strategy was used, inducing protective immunity through CD4+ and CD8+ T-cell-mediated responses [86].

The investigation of mRNA-based vaccines against AMR bacteria has just begun, and deeper studies on alternative routes of administration must be carried out. For example, the oral delivery of mRNA vaccines would enhance the mucosal response in response to the diffused presence of gut-associated lymphoid tissues (GALT). The mechanisms through which mucosal vaccines could elicit vaccine efficacy will be reviewed more deeply in the following chapters. Mucosal delivery, in fact, can limit infections caused by respiratory and digestive pathogens. For example, the exploitation of alphaviral replicon and salmonella bactofection for mRNA amplification are now being tested in the context of the development of new vaccines against SARS-CoV-2. Initially, the salmonella vector backbone was deeply modified to enable transcription in host cells and plasmid maintenance in bacteria. The study showed that immunisation via the oral route resulted in strong immune responses and cross-protective neutralising antibodies. It provided effective protection against SARS-CoV-2, including the B.1.617.2 Delta variant, in the lungs and nasal cavity of hamsters. The induction of sIgA likely played a role in the superior protection achieved via oral immunisation [87,88]. These promising results, obtained by exploiting the oral administration route, suggest the plausible applicability of mucosal administration of future mRNA-based vaccines protecting against AMR pathogens.

The COVID-19 pandemic in 2019 was the beginning of the most explosive vaccine development in history [89]. From this unprecedented competition, mRNA vaccines set themselves apart as the best alternative to traditional vaccine approaches in terms of low-cost manufacturing advantages and the incredibly efficient and fast development [90]. The results of these two different preclinical studies are the first examples for the effective use of an mRNA-based vaccine against an AMR pathogen, providing a new strategy for the rapid development of vaccines against potential future bacterial outbreaks and against pathogens already known to present AMR.

Table 1 provides a schematic overview for the antigen platforms used against some examples of AMR pathogens, indicating some examples of on-going clinical trials.

### 2.6. Future Perspective of Nano and Micro Vaccines Drug Delivery Systems (DDSs) and Their Application against AMR Pathogens

Many particulate antigen-delivery systems are currently under development. Typically, their size varies from the nano- to the micrometre range, which is similar to microbial pathogens, thereby potentially mimicking the immune system activation mechanism caused by a bacterial infection. Despite not always being immunogenic, nano- and microparticles may play a fundamental role in protecting the antigen from degradation and in promoting its efficient presentation to the immune system.

The Food and Drug Administration (FDA) defines nanoparticles among biological products as systems manufactured to have a size under 1000 nm [101], while microparticles are particles between 1 and 1000 μm in size [102]. In this review, some examples of nanosystems have been already discussed in the chapter related to vesicle-based antigens. This chapter will focus on the description of some examples of other delivery systems that are characterised by nano or micro dimensions that can be co-formulated with antigens and their subsequent impact on vaccine efficacy.

Nanoplatforms can be obtained from a large variety of synthetic or organic materials [101,102,103]. A clear physical advantage of nano-sized platforms is their capacity to easily move across body tissues. In fact, nano-sized particles can flow in the circulatory system, and particles in the range between 20 and 200 nm are able to reach the lymphatic system within a few hours of entering the body [104]. A second physical characteristic of DDSs that can improve antigen immunogenicity is the increase in size related to the antigen-presenting system: particles with a diameter of several microns can more easily be phagocytosed, leading to enhanced delivery of antigens to antigen-presenting cells (APCs) [105,106]. In the next part, we will describe novel DDSs, and discuss their advantages and applications to AMR pathogens.

#### 2.6.1. Self-Assembling Protein Nanoparticles (SAPNs) for Vaccine Delivery

The first reported example of a nano-sized platform for carrying antigens was self-assembling protein nanoparticles (SAPNs). SAPNs are macromolecular protein subunits that self-assemble in highly ordered structures called virus-like particles (VLP), similar to viral capsids, with a size of up to 200 nm. A wide variety of proteins can be exploited for the design of SAPNs; we here discuss some examples.

Currently, the most direct SAPN design method is the top-down adaptation of an existing protein naturally evolved from a viral capsid or cells. Typically, SAPNs are made up of many copies of monomers that, due to their arrangement in a symmetrical manner, form a closed, three-dimensional structure. One or several structural proteins can form a SAPN with the ability to self-assemble after their expression. VLPs, which closely resemble intact virions, were initially obtained by self-assembly of hepatitis B (HBV) or hepatitis P (HPV) virus capsid proteins. The bacteriophages Qβ and AP205, typically expressed in *E. coli* or ferritin protein, are further examples of SAPNs which have been used in the development of vaccines against bacteria [107]. Like viral structures, these molecules can directly activate the immune system via direct uptake from APCs and the presentation of the peptide to CD4+ T- and B-cells [108,109]. They maintain a high compatibility with biological systems and do not carry toxicity concerns due to their non-biodegradability [103]. Nanoparticle scaffolds deliver the antigen via genetic fusion or chemical or protein–peptide conjugation. Chemical conjugation requires modifications to some chemical groups on the VLP surface (such as hydroxyl, sulfhydryl, amino, or carboxylic groups); however, it is not always possible to achieve full modification of VLP monomers via chemical modification. Genetic modification is an alternative strategy based on the manipulation of structural viral genes to efficiently introduce the desired epitopes directly within the SAPN structure [110].

Despite the fact that SAPNs are currently mainly exploited for vaccination against viruses, there are some encouraging preclinical experiments that are investigating the use of protein nanoparticle in vaccines against AMR bacteria. A first example tested the fusion of βbarrel of fHbp with six SAPNs: ferritin, mI3, encapsulin, CP3, Qβ, and HbcAg NPs. This study demonstrated that it was possible to use genetic fusion to obtain homogeneous and well-structured molecules from all the particles with the exception of Qβ [111]. A recent study demonstrated the feasibility of co-expressing a *Helicobacter pylori* ferritin nanoparticle with the surface-exposed loop protein MtrE from *N. gonorrhoeae*, expanding the feasibility of this platform for vaccines against bacteria [112]. Furthermore, AP205 and Qβ were also recently tested for use in a vaccine against *S. aureus* [107]. Protection against infection was provided by both the SAPNs carrying the Hla antigen and was evaluated by measuring the induction of IgG antibodies and the protection against subcutaneous infection in immunised mice. Both the nanoparticles provided a significant IgG titre response and a reduction of the size of skin lesions, providing encouraging results for the exploitation of these platforms not only against viral but also AMR pathogen infections [113]. In a second example, a synthetic carbohydrate anti-*Salmonella enteritidis* vaccine was conjugated with bacteriophage Qβ. This vaccine was able to induce high levels of specific and long-lasting anti-glycan IgG antibodies protecting mice from lethal bacterial challenges in a passive transfer model [114].

The approval of vaccines against AMR pathogens using SAPN platforms remains a distant target, but their self-adjuvaticity and the possibility of carrying different antigens make them an intriguing potential carrier system for the rapid development of multiple vaccines. As mentioned earlier, the rapid development of vaccines against resistant pathogens is crucial for the containment of new epidemics. The use of platforms that do not require mixing with adjuvants further facilitates the formulation, development, and associated timelines for vaccine development.

#### 2.6.2. PLGA Based Nano and Microparticles DDS for Vaccines Delivery

Other promising nano- and microdelivery systems have been obtained using synthetic biodegradable polymers such as poly(lactic-co-glycolic acid) (PLGA). PLGA is regarded as biocompatible (it has been approved by the FDA and European Medicines Agency (EMA) for parenteral administration) and has high tolerance values for both intramuscular and subcutaneous injections [115]. In vaccine development, PLGA has been used to encapsulate antigens and then release them in a controlled manner. Recent studies of an HIV protein antigen, gp120, demonstrated the possibility of inducing a strong and long-lasting immune response after the administration of multiple injections or osmotic pumps with a constant release rate in both mice and primates [116].

PLGA micro- or nanoparticles can modulate the release of encapsulated molecules according to the rate of degradation of their matrix. The rate of particle degradation via monomer hydrolysis is strongly related to both the physical characteristics of the particle and the chemical characteristics of the polymer.

For PLGA to become more widely used within vaccine formulations, there are several points of consideration regarding its use from a clinical perspective that must be addressed. One of the main obstacles is the instability of the antigen during formulation when encapsulated inside the particles. Antigen denaturation can be induced during manufacture, in which organic solvents are typically used in emulsions with an aqueous phase. Moreover, denaturation of the encapsulated antigen can be also triggered by the acidification that takes place upon PLGA hydrolysis [115,117]. The use of buffering agents such as MgCO_3_ or Mg(OH)_2_ can be used to avoid an acidic environment [118].

Another issue hindering the increased use of microparticle formulations within the field of vaccine development is their inability to be sterilised. They are incompatible with widely used sterile filtration because of particle size and with irradiation systems due to antigen stability issues [119].

To overcome the issues related to degradation and inability to be sterilised, the loading of the antigen into pre-formed, sterilised, porous PLGA microparticles has shown promising results [120,121].

The adjuvanting effect of PLGA particles is related to their physical characteristics. Micro-sized PLGA particles, when injected intramuscularly or subcutaneously, induce depot formation, stimulating macrophages and inducing uptake of dendritic cells, which may therefore carry the antigens to the lymphatic system [122]. Conversely, when PLGA nanoparticles are smaller than 10 µm, they can pass the barrier of the lymphatic system to be directly transported to the lymph nodes after injection [122].

One method of increasing the immunogenicity of PLGA is the encapsulation of adjuvants directly within the nano- or microparticles. In two separate reports, antigens co-formulated with adjuvant were encapsulated in PLGA NPs.

Antigen encapsulation in PLGA particles does not depend on the nature of the encapsulated antigen, and its only limitation is its stability during the loading procedure and the release from polymeric matrix. Multiple antigens could potentially be contemporaneously encapsulated in a single microparticle, generating a self-adjuvated, multivalent antigen, with the possibility of inducing protection against multiple strains or different AMR pathogens.

No formulation exploiting PLGA-based nano- and microparticles for vaccine delivery has yet been approved. It is an intense area of research, however, with efforts ongoing towards the development of this platform within the context of antigen delivery. Here we report some examples of the most promising results obtained from particles used as adjuvant systems or antigen delivery platforms for vaccines against AMR pathogens.

The first reported PLGA NP encapsulating systems were cationic surfactants, methyl-dioctadecylammonium (DDA) bromide, and the immunopotentiator glycolipid trehalose-6,6′-dibehenate (TDB). They were used to immunise CB6F1 mice with major outer-membrane protein (MOMP) from *Chlamydia trachomatis*. The results obtained in the preclinical experiments demonstrated the comparability of the results obtained from the PLGA-DDA-TDB NPs and MOMP vaccines adjuvated with CAF01 incorporating both DDA and TDB55 [69]. In the second report, a *Mycobacterium tuberculolsis* HspX/EsxS subunit vaccine was investigated. The monophosphoryl lipid A adjuvant was encapsulated in hybrid PLGA/DDA nanoparticles. The vaccine/adjuvant PLGA nanoparticles were shown to induce a stronger antibody titre and Th1 response in comparison to the non-adjuvated system when used in a mouse model [123]. Hybrid, chitosan-containing polymeric NPs have shown promise as a delivery system for mucosal immunisation with pneumococcal proteins. A preclinical study investigating the immune response of mice post-mucosal immunisation with a PspA4Pro antigen adsorbed within PLGA/chitosan hybrid NPs, an increased rate of survival in immunised mice was reported (an 83% increase with respect to mice receiving the non-encapsulated PspA4Pro). These results suggest the potential adjuvant activity of PLGA/chitosan NPs in stimulating systemic immune responses. The supposed mechanism of action is related to the maturation of DCs upregulating CD40, CD80, and CD86, triggered by the production of MHC II molecules [124]. Another report detailed an improved CD8 T-cell immune response after the prolonged release of *M. tuberculosis* antigen Mtb8.4 after a single administration. Moreover, the response was better than the one obtained from Mtb8.4 protein/adjuvant combination, following one or two immunisations [125].

The generation of an effective immune response following a single administration makes it possible to immunise a greater number of recipients more quickly. The resulting increase in the immunity of the general population is a fundamental factor when attempting to stop the diffusion of mutant strains of pathogens. For this reason, microparticles may represent a promising platform for the design of innovative vaccine formulations for the prevention of AMR pathogens pandemic scenarios.

In conclusion, modern vaccine formulation development is moving towards the use of multi-antigen vaccines that can induce effective immunisation against single or multiple pathogens. The development of multivalent prophylactic treatments is critical to delivering improved immune protection against AMR pathogens and the spread of new mutant strains. Moreover, systems such as GMMAs and MAPS have the potential to expedite the distribution of new vaccines.

## 3. Adjuvants for Improved Vaccines against Antimicrobial Resistance Pathogens

In some cases, the immunogenicity of the administered antigen or the immune response within the target population (i.e., elderly people or children) is not sufficient to initiate or to boost an effective immune response similar to that obtained from natural infection [126]. In such cases, adjuvants can be used to increase the efficacy of the vaccine via different mechanisms of action, such as (i) by increasing the antigen half-life once injected, (ii) by activating or maturing antigen presenting cells (APCs), (iii) by favouring their capacity to uptake the antigen, (iv) by inducing immunoregulatory cytokines activation, (v) by activating inflammasomes, or (vi) inducing local inflammatory activity [127,128]. These multiple mechanisms of action are important not only for the modulation of the immune response, but also to reduce the administered dose or number of doses [128].

Such a variety of mechanisms can be obtained from a range of adjuvants such as emulsions, microparticles, mineral salts, saponins, microbial components, cytokines, and liposomes. To increase the efficacy of a greater number of antigens, new and diverse adjuvants must be developed. Due to the diversity of AMR pathogens, a range of vaccines with different types of antigens are required. A range of different adjuvant mechanisms will therefore be necessary to complement these diverse antigens.

Ideally, an adjuvant should be free from side effects and characterised by an excellent safety and stability profile. Moreover, any given adjuvant should be compatible with a large variety of vaccine classes and components, be easy to manufacture, and have low raw material costs [129].

In this section, the main and new examples of vaccines adjuvants will be described, presenting some examples of their use in vaccines against AMR pathogens.

### 3.1. Aluminium Salts

#### 3.1.1. Generalities on Aluminium Salts

Aluminium salts were one of the first examples of vaccine adjuvants. They have been used for over a century and are still widely used in vaccines today [130]. Most aluminium-based adjuvants used in vaccines have complex stoichiometries based mainly on one of two classes: aluminium hydroxide (AH) and aluminium phosphate (AP). Their immune-stimulatory activity derives from the release of uric acid from damaged or necrotic cells, which in turn activates dendritic cells via the NALP3 inflammasome [126]. In addition to activation of the innate immune response, aluminium formulations generate a depot at the site of injection that ensures prolonged antigen release and therefore a longer-lasting immunoresponse as compared to non-adjuvanted products [130]. Antigen adsorption to aluminium salts (AH and AP) mainly happens via electrostatic interactions between the aluminium and the antigens, which is driven by the point of zero charge (PZC) of the aluminium salt, the isoelectric point of the antigen, and the pH of the formulation buffer [131].

When considering the development of an aluminium-containing multicomponent vaccine, extensive pre-formulation and formulation studies of all the components are needed. These studies mainly investigate the effective adsorption of the antigenic components onto the aluminium [132]. In the case of broad-coverage bacterial vaccines with a multitude of antigens, those studies often increase in complexity as the operating windows of condition in which all antigens are stable and adsorbed on the aluminium becomes narrower with an increase in the number of antigens.

Despite their wide and longstanding use, aluminium-based adjuvants have several associated drawbacks. For example, they are very sensitive to freezing conditions that may occur during vaccine transportation or storage, which can cause the bond between the aluminium and the antigen to break, leading to an impairment of the immunogenicity [133].

One way of increasing the activity of aluminium adjuvants is to reduce the size of the aluminium particles to below the micro range. Preclinical studies performed on mouse models have highlighted a stronger immune response when mice were immunised with ovalbumin adjuvanted with AH nanoparticles with a size of ca. 100 nm as compared to traditional (micro-sized) AH [134]. Moreover, the so-called ‘nanoalum’ reduced inflammation at the site of injection due to the absence of an adjuvant depot while antigen uptake via APCs was still possible, stimulating a Th1 immune response [131,135]. Another formulative advantage deriving from the exploitation of nano-sized aluminium salts is the increase of the adjuvant surface area available to adsorb the antigen. In an optimised system, this would permit a lower amount of adjuvant to be used to administer the same dose of vaccine [131].

#### 3.1.2. Use of Aluminium-Based Adjuvants in Vaccines against AMR Pathogens

Aluminium is used as adjuvant in various vaccines against AMR pathogens, and some of them have been mentioned previously in this review. For example, the glycoconjugate-based vaccine Prevnar 20, which contains saccharides from 20 serotypes of *S. pneumoniae*, is formulated with aluminium phosphate. Aluminium hydroxide was also used in two of the already-mentioned vaccines against *Shigella*: the quadrivalent bioconjugate vaccine Shigella4V and the GMMA based vaccine 1790GAHB. These examples contain multiple antigens in a single formulation to protect against different pathogen serotypes, highlighting the effective use of aluminium adjuvants in formulation of vaccines against AMR pathogens.

### 3.2. Oligonucleotide Adjuvants

#### 3.2.1. Generalities on Oligonucleotides Adjuvants

There is significant interest in the discovery of safer and more effective adjuvants for vaccines. For this reason, new immune-modulating molecules have been explored, and a considerable improvement was obtained after the discovery of toll-like receptors (TLR) and their capacity to induce both adaptive and innate immune responses [136].

Many conserved bacterial structures can be recognised by TLRs, one of which is typical nucleic acid sequences, particularly bacteria present in their genome CpG motifs which are central unmethylated cytosine–guanine dinucleotides sequences plus flanking regions. To mimic the immunostimulatory effect obtained from CpG bacterial sequences, synthetic oligonucleotides with CpG sequences are used [137].

The commercial HEPLISAV-B vaccine (presented as a fully liquid formulation containing a CpG adjuvant), which is used against HBV, has demonstrated a good stability profile at 2–8 °C across its intended shelf life [138].

CpG oligodeoxynucleotide (ODN) adjuvants can be divided into different classes: A, B, C, P, and S, depending on the structural characteristics of their backbone and the immune response type they induce [139]. Furthermore, the tertiary structure of the different classes regulates their intracellular distribution, leading localisation in different endolysosomal compartments [140]. TLR9 stimulation induces the rapid activation of the innate immune system, supporting the adaptive immune response [139]. This mechanism induces an inflammatory response via the activation of related genes with the stimulation of *MyD88*, *IRAK*, and *TRAF-6*. The response is detectable within 30 min, with a peak after three hours that declines after three days [137]. In terms of a specific response, the CpG A-class induces APC maturation and pDC INF-α secretion; CpG B-class oligonucleotides can trigger strong B-cell and NK cell activation; and C-class CpGs stimulate the proliferation and differentiation of B-cells and plasmacytoid dendritic cells (pDCs) and trigger the production of interleukin-6 (IL-6) and interferon alpha (IFN-a) [137,141]. There is currently particular interest in the new P-class of ODNs that contain a double palindromic sequence and have been shown to induce a higher INF-α than C-class ODNs.

#### 3.2.2. Use of Oligonucleotides Adjuvants in Vaccines against AMR Pathogens

Although CpG adjuvants are mainly used in antiviral vaccines, new investigations regarding their efficacy in vaccines against *Pneumococcus* are currently ongoing. For example, synthetic ODNs have been demonstrated to be effective in promoting a higher antibody titre in mice receiving *S. pneumonie* glycoconjugate-based vaccine [142]. Phase 1a/2b clinical trials using CpG with a pneumococcal conjugate vaccine 7vPnC (Prevnar) and PPV-23 (Pneumo Novum) administered to human immunodeficiency virus (HIV)-positive patients demonstrated a significantly higher response in the group that received the B-class CpG-adjuvanted vaccines versus the control group [139,141].

These discoveries promote the exploitation of these adjuvants in the development of new vaccines against AMR pathogens. Moreover, the good results obtained from the co-administration of CpG adjuvants with already-approved 7vPnC (Prevnar) and PPV-23; (Pneumo Novum) vaccines encourage the rational refinement of already-existing vaccine formulations. In this way, a faster and cheaper development of new prophylactic treatments should become possible, avoiding the time and cost related to the discovery of new antigens.

### 3.3. Emulsion Adjuvants

#### 3.3.1. Generalities on Emulsion Adjuvants

An emulsion can be simply described as the dispersion of two immiscible liquid phases (e.g., a water phase and an oil phase) containing stabilisers and surfactants to increase dispersion stability [143]. Emulsion-based adjuvants represent an important component in many vaccine compositions. Oil-in-water (O/W) emulsions are often used due to their high tolerability and reduced viscosity [140]. Emulsion adjuvants are characterised by several typical mechanisms, such as the rapid transport of adjuvants from the muscle to the draining lymph nodes without any detectable depot effect, causing the strong, and synchronised activation of innate immune cells in the draining lymph nodes. Emulsion adjuvants also yield an increase of CD4+ T-cells, T-follicular helper cells (TFHs), and germinal centres (GCs), leading to an enhanced humoral immune response in terms of both quantity and quality. Finally, a complex and varied pattern of danger-associated molecular patterns (DAMPs) and mediated pattern recognition receptor (PRR) activation has also been observed [144].

The first oil-in-water emulsion adjuvant licensed for human use was developed by Chiron and is called MF59, which was used for the first time in 1997 in a seasonal influenza vaccine. MF59 is an emulsion adjuvant made of an oil-in-water emulsion in which the oily phase contains squalene. It forms vesicles with an average size of 160 nm with uniform morphology and high stability [145]. Another important example of an oil-in-water emulsion-based adjuvant is AS03, which was developed by GSK. It is composed of a mixture of DL-α- tocopherol, squalene, and polysorbate 80 [146]. AS03 and MF59 share many characteristics, but the additional presence of DL-α-tocopherol in AS03 increases the strength of the antigen-specific adaptive response, increasing cytokine production, and inducing the early migration of eosinophils and neutrophils to the draining lymph nodes as well as the loading of antigen in the monocytes [147,148]. Emulsion manufacturing depends heavily on the nature and concentration of surfactants. These compounds play a crucial role in reducing the interfacial tension between oil and water, allowing for the formation of stable O/W nano-emulsions. To achieve long-term stability, surfactant molecules must be flexible enough to conform to the high surface curvature of dispersed oil droplets, and their concentration should be sufficient to cover the entire surface area of the droplets. The selection of surfactant is carried out based on their hydrophilic/lipophilic balance [149]. Emulsion-based adjuvants can also be filtered through 0.22 µm membranes to guarantee the sterility of the formulation, and they can be stored at 2–8 °C for years [150].

#### 3.3.2. Use of Emulsion Adjuvants in Vaccines against AMR Pathogen

MF59 has so far only been approved for use in influenza vaccines; however, multiple clinical trials are now investigating its use in antimicrobial vaccines [151]. For example, a new phase 1 clinical trial is currently investigating the safety of a PCV-13 vaccine against pneumococcal disease when administered with MF59 as an adjuvant, concurrently with an MF59-adjuvated influenza vaccine in elderly people (age > 60) [152].

AS03 was used for the 2009 H1N1 influenza pandemic [146,153]. As with MF59, only a few clinical trials investigating the adjuvant effect of AS03 for bacterial vaccines have been performed so far [122]. Very recently, a new clinical trial testing the safety of AS03 with Kleb4V antigens (the predominant O antigen-polysaccharides of *Klebsiella pneumoniae*) was performed in adults (aged 18 to 40 years) and subsequently in the target population of older adults aged 55 to 70 years, but the data are yet to be published [154]. The results of these clinical trials are the first examples for the effective use of emulsion-based adjuvants in vaccines against an AMR pathogen.

### 3.4. Liposome-Based Adjuvants

#### 3.4.1. Generalities on Liposome-Based Adjuvant

In 1960, a biocompatible and biodegradable phospholipidic bilayer structure, known as a liposome, was developed for the first time [155]. Phospholipids are the main components of liposome bilayers and are made of a hydrophilic head and a hydrophobic tail. The hydrophilic heads are exposed on the external side of the liposome shell, while their hydrophobic tails make up the inner side of the bilayer structure [156]. These structures are considered one of the most effective and versatile drug delivery systems due to their high loading capacity, biocompatibility, and simple functionalisation of their surface [157]. Liposomes are usually distinguished as belonging to two major types, according to their morphology: unilamellar liposome vescicles (ULVs) due to the presence of a single phospholipidic bilayer around an aqueous core and multilamellar liposome vescicles (MLV), which are defined by multiple bilayers with a concentric structure separated by aqueous compartments [157].

There are several methods used to prepare liposomes, such as reverse-phase evaporation, thin-film hydration, detergent depletion, solvent injection, and the emulsion method [156,158,159]. The most recent techniques are based on microfluidic preparations that are highly controllable and reproducible [160]. Microfluidic systems exploit the mixing of very small volumes of two liquid phases, usually a water phase and an organic phase, inside intersecting microchannels. Liposome stability depends on several formulation parameters, such as osmolarity, salinity, pH, and temperature [160,161].

Liposomes can deliver both hydrophilic and hydrophobic molecules that are distributed inside the liposome nucleus or onto the membrane, respectively. For example, during the manufacture of liposomes, hydrophobic molecules can be dissolved into the lipid-containing organic phase and incorporated in the inner lipophilic phase of the liposome bilayer structure. Conversely, hydrophilic molecules are easily encapsulated in the hydrophilic core of liposome after dissolution in the aqueous phase [155].

One well-established liposomal adjuvant being used in bacterial vaccines already on the market and in vaccines currently undergoing clinical trials is AS01. This system contains two different immunostimulants: 3-O-desacyl–monophosphoryl lipid A (MPL) and the saponin QS-21 [162]. The MPL component triggers the activation of antigen-presenting cells (APC) via TLR4 activation and stimulates an antigen specific T-cell response [163]. The mechanism of immune system activation via QS-21 remains unclear; however, recent studies have suggested a possible interaction of QS-21 with innate pathways in monocytes, triggering IL-1β/IL-18 after Nlrp3 inflammasome activation [164]. As the AS01 system contains both MPL and QS-21 molecules, it is able to induce a very strong and synergistic adjuvant effect [162]. Some specific features related to the AS01 system distinguish it from other adjuvant systems: (i) no depot effect; (ii) the activation of a broad spectrum of APC populations; and (iii) the synergistic effect of MPL and QS21, which triggers a greater antigen-specific response. There are already examples of approved vaccines formulated with AS01, such as Shingrix (against shingles) and Mosquirix (against malaria), and others are currently in clinical trials.

Another liposome-based adjuvant platform was developed by the Walter Reed Army Institute of Research (WRAIR) in 1986. This Army Liposome Formulation (ALF) was characterised by liposomes containing lipid A, which was derived from Gram-negative bacterial lipopolysaccharide (LPS) [165]. All molecules composing ALF, and ALFs containing QS-21 (ALFQ), are amphiphilic, and the liposomal phospholipid bilayers are stabilised by van der Waals forces. In vitro experiments have demonstrated the importance of QS-21 in ALFQ formulations for the induction of IFN-γ, and both ALF and ALFQ induce both Th1 and Th2 responses, together with strong levels of IL-4 [166,167].

#### 3.4.2. Use of Liposome-Based Adjuvant in Vaccines against AMR Pathogens

The use of liposome-based adjuvant in vaccines against AMR pathogens is currently under investigation in several clinical trials against different antigens. For example, M72/AS01 is a subunit fusion protein derived from two *Mycobacterium tuberculosis* (Mtb) proteins, adjuvated with AS01. This vaccine finished phase 2b clinical trials (NCT01755598) with participants aged between 18 and 50 years old and has demonstrated up to 50% protection against active tuberculosis [15,168]. Moreover, GlaxoSmithKline (GSK) recently evaluated the safety and immunogenicity of an F2-based vaccine against *Clostridioides difficile* in a phase 1 trial. The vaccine with and without AS01 is administered to healthy adults aged 18 to 45 years and 50 to 70 years.

New liposome-based formulations with active synthetic monophosphoryl lipid A (phosphorylated hexacyl disaccharide, PHAD) are currently under development. A recent study investigating a vaccine against *E. coli*, testing a range of PHAD dosages (10, 20, 40, or 43 μg) with the antigen FimHC, was demonstrated to be safe and immunogenic at all doses when administered to women aged 21 to 64 [169].

In conclusion, the judicious use of the appropriate adjuvant in vaccines allows for a reduction in the number of injections required to achieve effective immunisation. This, in turn, enables a more rapid diffusion of immunity within the population, creating a barrier against the spread of new mutant strains while simultaneously slowing down the diffusion of diseases caused by antimicrobial-resistant pathogens. Table 2 provides a schematic overview for the adjuvants used in the formulation of vaccines against some AMR pathogens, indicating examples of on-going clinical trials.

## 4. Alternative Vaccines Administration Routes in AMR Vaccines

The traditional route of vaccine administration (i.e., injection via syringe) delivers a strong immunological response, but often confers low T-cell-mediated immunity and low mucosal protection [176]. Alternative routes of vaccine administration that can elicit improved responses are therefore of great interest. For example, immunisation via mucosal or intradermal administration can induce long term B-cell and T-cell memory and better antigen presentation to APCs, enabling strong local and systemic protection [176,177].

Mucosal tissue is composed of a follicle-associated epithelium that contains microfold cells. These are specialised for antigen endocytosis and can promote several immune mechanisms, such as the presentation of antigens to macrophages and DCs. This mechanism promotes an immune response localised in targeted mucosa by imprinting homing properties on T- and B-cells [178]. Moreover, M-cells favour the production of antigen-specific secretory IgA, which reduces pathogen adhesion to the epithelial cells.

When vaccines are delivered via mucosal tissues (such as oral, respiratory, rectal, or intravaginal), it is important to consider the low immunogenicity of soluble antigens in such systems. When mucosal sites are exposed to soluble antigens, a tolerance mechanism is naturally induced to prevent an excessive inflammatory response derived by exposure to various naturally presenting molecules. For this reason, the correct composition of the vaccine, via the selection of the appropriate combination of adjuvant and delivery system, is mandatory for the development of an effective mucosal vaccine. To overcome this challenge a biodegradable polymeric PLGA particle, which is able to shield antigens from digestive enzymes, has been widely acknowledged as an effective vehicle for delivering antigens via oral administration [179]. As with biodegradable polymeric particles, lipid-based nanoparticles can be exploited for the delivery of soluble antigens within mucosal tissues, as they are able to encapsulate both antigens and adjuvants [180].

There are already examples of existing oral vaccines that exploit this alternative route of administration, such as Vivotif (Crucell) and Ty21, which is a live attenuated vaccine that is administered orally. In a trial involving adults and children over the age of 6 against Salmonella typhi *Ty21a*, the efficacy of the product was demonstrated, albeit with a variability of up to 50%. This variability has been correlated with the different microflora of the recipients, their differing nutritional status, and their levels of pre-existing natural or maternal antibodies. This demonstrates a critical issue that has impeded the worldwide spread of mucosal vaccines, specifically the unfavourable outcome of these vaccines in developing countries, commonly known as the tropical barrier [181,182]. The safety of an intranasal vaccination within an adult population was investigated in a recent clinical trial of a vaccine against *Shigella flexneri*, Invaplex 50, a macromolecular complex containing IpaB, IpaC, and LPS, formulated from an aqueous extract of virulent *Shigella*. Promisingly, the trial demonstrated the capacity of the vaccine to elicit IgG production and highlighted good tolerance with minor short-lived nasal symptoms and no evidence of dose effect [183].

Another alternative administration route is via intradermal (ID) vaccine administration. With this approach, the antigen is delivered into the dermis, mainly targeting the dermal DCs and macrophages. Recent improvements have allowed for the development of new ID injection technologies such as microneedles (MN) [184]. MNs are usually engineered in ID patches composed of FDA-approved polymeric materials (e.g., polyvinyl alcohol (PVA), polyvinyl pyrrolidone (PVP), hyaluronic acid, and polylactic acid). These needles are micro-sized, which allows them to effectively transport a wide range of drugs and small particles whilst causing reduced discomfort to the recipient during their application compared to the use of a syringe [185].

Dose administration via skin patch was also shown to be effective in a phase 2 clinical trial for a vaccine preventing diarrhoea. In the trial, healthy adults (aged 18 to 64 years), were vaccinated with a heat-labile enterotoxin (LT), derived from Enterotoxigenic *E. coli* (ETEC) [186,187].

## 5. Can Antimicrobial Resistance Lead to A New Era for Vaccine Analytics?

With the increasing demand for new vaccines against existing and emerging infectious diseases, analytical strategies should explore new technologies and approaches in order to meet the highest standards for safety and efficacy within increasingly strict timelines. The International Council of Harmonisation (ICH) defines specific guidelines that conceptualise new quality paradigms for the development of new products based on scientific and risk-based approaches [188]. At the heart of the analytical control strategy is the detailed identification of the potential quality attributes of the vaccine product (from drug substance until final vaccine) that have, or could have, an impact on safety and efficacy [189]. Once an attribute is identified, a deep screening for the best analytical approach begins. The selected method(s) should be able to measure and monitor the attributes whilst thoroughly responding to the specific analytical parameters requested by every pharmacopeia, such as specificity; precision; accuracy; linearity range; and LOD/LOQ [190].

Most current analytical control strategies are focused on the traditional “one CQA—one analytical method” approach. With the acceleration of clinical trials, however, products will have to be developed and released more quickly, with a consistent package of high-quality data. A shift from the traditional model towards a more innovative way of working will therefore be required to ensure this transition. Covering more attributes in the same analytical session would allow for more product information to be accumulated more quickly, saving time and increasing efficiency. The concept of multiple-attribute monitoring (MAM) is already well established [191], and in the recent years, the improvement of multiple cutting-edge analytical technologies has also allowed for the application of this concept to analytical fields that were previously restricted only to highly specialised laboratories [192,193].

Mass spectrometry (MS) plays a pivotal role in vaccine analytics [194] and is perfectly suited to the concept of MAM, since many attributes can be monitored within the same analytical session. These attributes are not limited only to the antigen, but can include all components of a vaccine formulation, such as the adjuvant and other excipients. If used to its full potential, MS is able to give detailed information on, amongst other things: concentration; molecular weight; identity; and biochemical modifications (such as post translational modifications (PTMs)). For example, the absolute quantification of protein content, asparagine deamidation, and host cell proteins in the same analytical session has recently been reported. A coulometric mass-spectrometry approach, based on the electrochemical properties of peptides, was used without the need for external standards or calibration curves [195]. Furthermore, the unprecedented sensitivity (up to the femtomolar range) and the dynamic range of linearity [196] achieved by the most recent generation of mass spectrometers can offer unique applications in the field of vaccine characterisation. The simple coupling of MS with different separative techniques (e.g., liquid chromatography or capillary electrophoresis) is well known, and has allowed for the simultaneous monitoring of several multicomponent vaccines [197]. MS can also facilitate other analytical approaches: critical immunoassay reagents used for antigeniticy and potency studies, such as monoclonal antibodies, can be characterised by hydrogen-deuterium exchange coupled with MS in order to define the epitope [198,199,200].

Other techniques can also be applied within the MAM concept. Recently, size-exclusion chromatography (SEC) has been employed to monitor different CQAs in a novel pneumococcal vaccine in a single chromatography run. The developed method was able to monitor the polysaccharide (PS) and protein (Pr) contents and the PS/Pr ratio, in addition to the molecular weight (Mw). Inline dynamic light scattering (DLS) and viscometry detectors were applied to further characterise the product, providing information on the particle size and conformation [201].

Nuclear magnetic resonance (NMR) spectroscopy plays a predominant role in the field of polisaccharide and glyconjugate vaccines [202,203,204]. This non-destructive technique can be implemented to confirm the structural identity and stereoisomerism of polysaccharide antigens, to quantitate decorative groups (e.g., O-acetyl [205]), and to measure polysaccharide content. NMR can also be used to quantify process- (e.g., CTAB, DOC [206], ethanol, cholesterol [207]) and product-related impurities (e.g., cell-wall capsular polysaccharide [208,209]). This technology is faster and produces higher-quality results as compared to more traditional methods such as colorimetric assays. As with mass spectrometry, NMR can also complement biochemical information for immunological tools, and several carbohydrate-based antigen epitopes have been mapped over the past decades. Saturation transfer difference NMR (STD-NMR) has been used to map epitopes more quickly, and with a higher level of detail, than X-ray diffraction, which is time-consuming and challenging to undertake [210].

Even if the major limitation for NMR is sensitivity, this aspect has been already improved through improvements in hardware, magnetic field strength, probes design and pulse sequence programs [211]. Thus, the last frontier for NMR will be the characterisation of glycan structures present in the complex matrix of multivalent vaccine formulations, which will be surely accomplished when the latest generation of high-field NMR spectrometers are used not only in discovery labs but also in biopharmaceutical facilities [212,213].

The last step will be the translation of all these analytical innovations in the routine analysis of commercial vaccines, moving the newly developed tools into quality control laboratories. NMR and MS have recently entered into QC environments thanks to improved instrument robustness and GMP compliant software (such as CFR 21 part 11, and others), confirming that a new wave in vaccine analytics has already started.

In the near future, the involvement of machine learning (ML) [214] and artificial intelligence (AI) [215] will be instrumental in selecting the correct critical quality attributes, and predictive models will enable the evaluation of vaccine stability or the interconnection between biochemical features and potency. Looking back to the 2000s, the first attempt towards the use of bioinformatics to revolutionise vaccine discovery was achieved by Rino Rappuoli, who exploited the advances in genome-sequencing technology to develop reverse vaccinology [216]. This was then used for the identification of several surface-expressed antigens of *Neisseria meningitis* B (MenB). This approach enabled the identification of the antigens within the first vaccine against MenB infections [217]. Twenty years later, several ML software programmes have been developed to identify vaccine candidates, enabling the prediction of suitable bacterial antigens also in the contest of the vaccine formulation [218,219,220,221].

A special reflection should be undertaken for the evaluation of vaccine potency [222]. This attribute is usually carefully monitored, since it plays a central role as an indicator of an antigen’s biological efficacy and as validation of effective antigen design [223]. Potency, whether it is measured by an in vitro or an in vivo assay, is, by definition, a quantifiable biological response elicited by the antigen as a drug substance (DS) or when formulated in the vaccine product [223]. A dose is a quantitative measure of the active content, usually the antigen or the precursor of the antigen (e.g., mRNA). To date, one of most complex measurements is the evaluation of antigen content within vaccine formulations, which can include different biological molecules with several variants and the concomitant presence of other formulation components (e.g., excipients) and adjuvants. In this context, antibody-based immunoassays offer the possibility of correlating in vitro potency of the antigen (also called antigenicity) with its content [222].

In conclusion, this new way of analysing vaccines will help shorten the research and development time of novel vaccines, representing a fundamental step in the fast-changing scenarios of AMR.

## 6. Conclusions

AMR is an urgent and global threat to human health, and vaccination can be used as a primary tool to prevent the spread of bacterial infections and reduce the use of antibiotics. To do so, there is a continuing need to develop vaccines with increased potency and broader coverage.

Developing a vaccine is a long and costly process with a high failure rate. As discussed within this article, a significant portion of vaccine-candidates against AMR pathogens are multivalent, and also include other components that vastly differ in terms of their structural features (e.g., excipients, adjuvants). This variety often results in rather narrow, and therefore complex, development ranges for formulation. The development of stable vaccine formulations should consider the inherently complex structure of each antigenic component (e.g., proteins, glycoconjugate, bioconjugates, etc.) and the multiple degradation pathways these biological structures may undergo. Extensive pre-formulation studies should provide a solid scientific foundation to enable formulation development with the aim of better understanding the properties of antigens and adjuvants and of characterising the impact of their combination on product features.

Complex vaccine formulations, compriing multiple and diverse antigens with appropriate adjuvants and excipients, delivered in the correct manner, will be required to properly address the problem of AMR. Some examples of such vaccines have already been on the market for several years (e.g., Boostrix^®^, approved in 2011 by FDA for the prevention of tetanus, diphtheria, and pertussis; Prevnar 20^®^, approved by FDA in 2021, for the prevention of 20 types of pneumococcal bacteria).

In the future, to increase the breadth of protection offered by vaccines against bacterial targets, to the use of technologies that are inherently “multivalent friendly”, such as the MAPS, might be preferentially considered. Another window of opportunity could lie in mRNA vaccines, although examples of bacterial mRNA vaccines currently scarce, their success in the field of viral vaccination could lead to mRNA vaccines against bacterial pathogens.

Vaccines can be used not only to prevent outbreaks of AMR, but also to help reduce the burden associated with bacteria having already acquired AMR. A feature of AMR is the speed with which an outbreak might occur, which means scientists worldwide must always be alert and reactive to potential signs of AMR. Much like in a pandemic setting, the time required to develop a vaccine that was not needed a year ago becomes key. The recent pandemic has shown that the technology needed to be ready, but also that the maturity of product knowledge and the ability to develop appropriate analytical strategies was key.

## Figures and Tables

**Figure 1 ijms-24-12054-f001:**
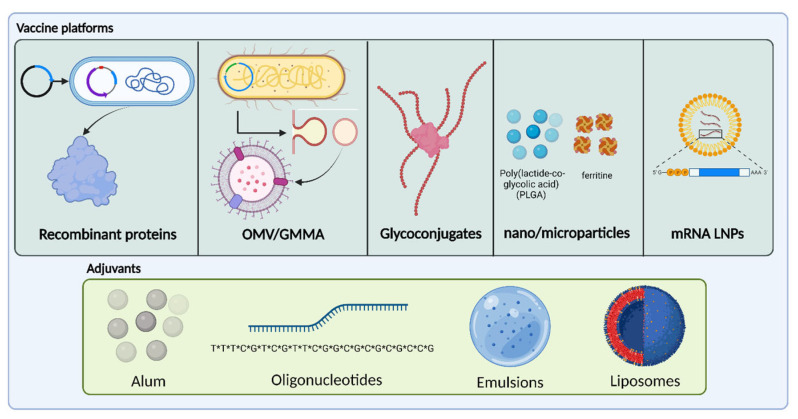
Schematic representation of the vaccines platforms and adjuvants present in most of AMR vaccines formulations. (*) indicate the phosphorothioate bonds. Created with BioRender.com.

**Figure 2 ijms-24-12054-f002:**
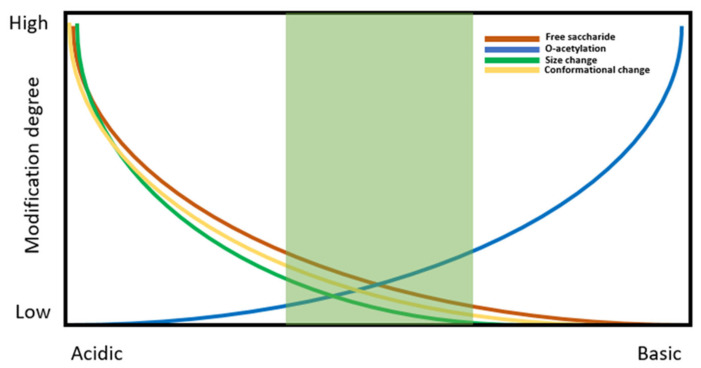
Schematic representation of formulation buffer pH impact on protein structure and stability.

**Table 1 ijms-24-12054-t001:** Schematic representation of the pathogens treated with the described platforms and examples of on-going clinical trials.

Platform	Pathogen	Clinical Trial
Recombinant protein antigens	*Streptococcus pneumoniae*	phase 2 [91]
	*Clostridioides difficile*	phase 1 [92]
	*Staphylococcus aureus*	phase 1 [93]
Glycoconjugate/bioconjugate	*Shigella flexneri*	phase2-phase 2b [94,95]
	*Streptococcus pneumoniae*	phase 1 [96]
	*Escherichia coli*	phase 1/2 [97]
OMV/GMMA	*Escherichia coli*	phase 2 [98]
	*Shigella sonneii*	phase 2 [99]
	*Shigella flexneri*	phase 1/2 [100]
RNA	*Yersinia pestis*	preclinical [83]
	*P. aeruginosa*	preclinical [85]
	*GAS/GBS*	preclinical [84]
	*Mycobacterium tuberculosis*	preclinical [86]

**Table 2 ijms-24-12054-t002:** Summary of main adjuvants investigated for bacterial vaccines with their supposed mode of action and associated clinical trials phase.

Adjuvant	Mechanism of Action	Pathogen	Antigen	Clinical Trial
Alum	Activation of dendritic cells via Nlrp3 inflammasome and induction of innate immune response via interaction between CD11c+ dendritic cells and lymphocytes T [130].	*Staphylococcus aureus*	Recombinant protein	phase 1 [93]
		*Streptococcus pneumoniae*	Recombinant protein	phase 1\2 [170]
		*Campylobacter* spp.	glycoconjugate	phase 1 [171]
		*Shigella flexneri*	bioconjugate	phase 2b [95]
CpG	Stimulation of TLR9 causing the induction of MyD88 pathway and type I interferon [137].	*Mycobacterium tuberculosis*	Recombinant protein	Phase 3 [172]
AS01	Stimulation of T-cell and TLR4, activation of IL-1β/IL-18 after Nlrp3 inflammasome interaction [164].	*Mycobacterium tuberculosis* *Clostridioides difficiles*	Recombinant protein Recombinant protein	phase 2b [168] phase 1 [92]
MF59/AS03	Induction of CD4+ T-cells, T-follicular helper cells (TFHs), and germinal centres (GCs); TLR9-independent activation of MyD88 [173,174].	*Streptococcus pneumoniae* *Klebsiella pneumoniae*	Polysaccharide bioconjugate	phase 1 [175] phase 1 [154]

## Data Availability

No new data were created in this review.

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
