# Peer review of "Evolution of Vaccines Formulation to Tackle the Challenge of Anti-Microbial Resistant Pathogens"

_ijms, 2023, doi:10.3390/ijms241512054_

Round 1

Reviewer 1 Report

The review article by Tognetti et al gives a comprehensive overview on different vaccination modalities to tackle anti-microbial resistance across bacterial species. The article is very well written and nicely structured into different sub sections.

Please find below my minor comments:

1. While describing mRNA-LNP based vaccines against bacterial infections such as Yersinia and Pseudomonas, the authors could consider including a short section on recent strategies of oral delivery of mRNA-LNP vaccines against bacterial pathogens such as Salmonella by employing self-amplifying RNA replicons from alphaviruses. As a reference, please refer to the article below:

https://www.frontiersin.org/articles/10.3389/fimmu.2022.884862/full#B23

2.      Along similar lines, the authors could consider including a section on novel approaches for vaccine against Mycobacterium using a viral replicon RNA (repRNA) formulated in nanostructured lipid carrier. Please refer to the article below: https://pubmed.ncbi.nlm.nih.gov/36679975/

Author Response

Daniela Stranges, PhD

GSK Vaccines S.r.l

Via Fiorentina 1, I-53100 Siena, ITALY

Tel: +39 3442996838

E-mail: daniela.x.stranges@gsk.com

July 12th, 2023

Submission of revised manuscript

Thank you very much for your support in the reviewing process of our manuscript entitled: ‘Evolution of vaccines formulation to tackle the challenge of anti-microbial resistant pathogens’.

We firstly acknowledge all the Reviewers for the very deep revision.

All the issues raised by Reviewer #1 have been addressed as detailed below.

Reviewer 1:

We thank the reviewer for the suggestions, and we included both references in the RNA-section.

We believe that it the present form the manuscript is acceptable for publication and look forward to a positive response.

Many thanks in advance for your support and consideration.

Yours sincerely,

Daniela Stranges, PhD

Reviewer 2 Report

Dear authors,

The manuscript "Evolution of vaccine formulation to tackle the challenge of anti-microbial resistant pathogens" is about an extremely relevant topic. The development of vaccines to prevent diseases caused by microorganisms resistant to antimicrobials is a priority.  And in this context, literature reviews are extremely important for the dissemination of scientific information.  Therefore, reviews must be very precise regarding the use of bibliographic references.

References must be the source of information cited in the text. It is essential for the reader to have access and be able to analyze how this information was obtained.

In this regard, the manuscript in the form presented does not achieve this objective.

An analysis of the first ten references used shows that only two are directly related to the information contained in the text (references 3 and 9).

For example, the information "The discovery of chemically synthetized antimicrobials in the 19th century was a milestone in the field of medical research, and in 1928 Alexander Fleming introduced the world to a new era of bacterial disease treatment when he identified the first non toxic antibiotic (1.e., Penicillin G)[1]." is not described in Ref1. The information "Since then, the combination of antimicrobials and antibiotics has contributed to the prolongment of human life expectancy, and people in high-income countries can now expect to live to 81 years of age[2]."is not described in Ref 2.

In addition, the text provides incomplete information (see ref 10). In the text, the authors inform "In 2018 an expert pool on behalf of Global Alliance for Vaccines and Immunization (GAVI-The Vaccines alliance) defined the selection criteria for vaccines against AMR pathogens: reduction of mortality, prevention of use of antibiotics, morbidity, sense of urgency due to treatment impact and ethical importance[10]." but do not mention the "Societal impact".

Therefore, I suggest that the authors make a detailed review of the cited references.

Author Response

Daniela Stranges, PhD

GSK Vaccines S.r.l

Via Fiorentina 1, I-53100 Siena, ITALY

Tel: +39 3442996838

E-mail: daniela.x.stranges@gsk.com

July 12th, 2023

Submission of revised manuscript

Thank you very much for your support in the reviewing process of our manuscript entitled: ‘Evolution of vaccines formulation to tackle the challenge of anti-microbial resistant pathogens’.

We firstly acknowledge all the Reviewers for the very deep revision.

All the issues raised by Reviewer #2 have been addressed as detailed below.

Reviewer 2:

We acknowledge the reviewer that bibliographic references are the cornerstone of scientific communication. We confirm that there has been an error while combining versions that caused disruption in the bibliography in the introductory section, which led to references not corresponding to the citations.

We fixed this mistake and reviewed carefully all the citations in the whole review.

We believe that it the present form the manuscript is acceptable for publication and look forward to a positive response.

Many thanks in advance for your support and consideration.

Yours sincerely,

Daniela Stranges, PhD

Round 2

Reviewer 2 Report

Dear Authors,

The modifications made to the manuscript made it suitable for contributing to the development of vaccines against microorganisms.